# Telehealth Reduces Missed Appointments in Pediatric Patients with Tuberculosis Infection

**DOI:** 10.3390/tropicalmed7020026

**Published:** 2022-02-14

**Authors:** Angela Zhao, Nirali Butala, Casey Morgan Luc, Richard Feinn, Thomas S. Murray

**Affiliations:** 1Yale University, New Haven, CT 06510, USA; angela.zhao@yale.edu; 2Department of Pediatrics, Yale School of Medicine, New Haven, CT 06510, USA; nirali.butala@yale.edu; 3Department of Epidemiology of Microbial Diseases, Yale School of Public Health, New Haven, CT 06510, USA; cluc2@uic.edu; 4Department of Medical Sciences, Frank H. Netter School of Medicine at Quinnipiac, Hamden, CT 06473, USA; richard.feinn@quinnipiac.edu; 5Section Infectious Diseases and Global Health, Department of Pediatrics, Yale School of Medicine, New Haven, CT 06510, USA

**Keywords:** tuberculosis infection, missed appointments, COVID-19, telehealth

## Abstract

Pediatric patients with untreated tuberculosis infection (TBI), also called latent TBI, are at risk of progression to active TB disease. The primary aim of this study was to identify factors associated with higher rates of missed appointments and failure to complete therapy for pediatric patients with TBI. A secondary aim was to determine the impact of the COVID-19 pandemic and the rise of telehealth on TBI missed appointment rates. We first performed a retrospective chart review of 129 pediatric patients referred to the free Yale Pediatric Winchester Chest Tuberculosis Clinic from 2016–2019. Associations between demographic/clinical variables and missed appointments/failure to complete therapy were analyzed using univariate and bivariate chi-square tests. Language, lack of primary provider, and distance to clinic were the main contributors to missed appointments and poor treatment adherence. There was an association between the number of missed appointments and failure to complete treatment (*p* = 0.050). A second cohort of 29 patients was analyzed from January–December 2021 when telehealth was offered for follow-up appointments. Of these follow-up visits, 54% were conducted via telehealth, and the clinic’s missed appointment rate dropped significantly from 16.9% to 5.8% during this time frame (*p* = 0.037). These data demonstrate that telehealth is accepted as an alternative by patients for follow-up TBI visits.

## 1. Introduction

Approximately one-fourth of the global population is infected with tuberculosis (TB) [1]. TB infection (TBI) or latent TB is defined by an immune response against *Mycobacterium tuberculosis* in individuals without clinical or microbial evidence of active TB disease. While most healthy individuals with TBI never develop active TB, 5–15% progress to active disease after a variable period of latency, with the highest risk within 2 years of initial infection [2]. Younger children have a higher risk for progression to active disease compared with the general population, highlighting the importance of treating children with TBI [3,4]. The management and treatment of TBI is a crucial strategic component of TB prevention and control, as over 80% of active TB cases result from the reactivation of TBI [5].

Many studies have addressed challenges that TBI patients face in completing treatment. Previous literature has identified language, the severity of isoniazid and rifampin adverse effects, and transportation as barriers to treatment completion [6,7,8]. Treatment duration has also been identified as a factor associated with treatment completion; studies have shown that TBI patients were more likely to complete treatment using a 4-month regimen of daily rifampin (4R) compared with a 9-month isoniazid regimen (9H) [9,10]. However, less is known about specific factors associated with missed appointments that lead to failed therapy and the effect of the SARS-CoV-2, COVID-19 pandemic on the delivery of care for children with TBI.

The COVID-19 pandemic has presented healthcare systems globally with unprecedented obstacles, including the safe care of populations while reducing the risk of SARS-CoV-2 transmission. For example, there has been a decrease in the identification of patients with TB and an increase in TB-related mortality [11]. One approach to improve access to care is the implementation of telehealth services in hospitals and clinics [12]. Telehealth services allow for patients to overcome physical barriers to access medical care, promote continuity of care, and improve patient adherence [13]. One prospective study demonstrated that patients at a primary and specialty care clinic had a significantly lower no show rate for telehealth visits compared with no show rates for in-office visits both before and during the pandemic [14]. Although this adoption of telehealth is still nascent and actively evolving based on the needs of patients and health system resources, there is a history of telehealth use for video-based direct observed therapy (vDOT) for patients with active TB disease [15,16]. VDOT is a convenient and cost-effective alternative to daily in-person observation that is especially valuable during the COVID-19 pandemic [17]. Whether this benefit extends to appointments for children undergoing daily TBI therapy that is not directly observed has not been well studied.

The primary objective of this study was to identify factors associated with higher rates of missed appointments leading to incomplete therapy in TBI pediatric patients. A secondary objective was to study the impact of telehealth introduced into the clinic during the COVID-19 pandemic on missed appointments and completion of TBI therapy.

## 2. Materials and Methods

### 2.1. Settings/Patients

Yale New Haven Children’s Hospital Winchester Chest Pediatric Tuberculosis Clinic is a free clinic in New Haven, Connecticut in the Northeastern United States that is a regional referral center for children who require therapy for either TBI or for active TB disease. The clinic receives referrals from physician offices, public health departments, school-based health centers, refugee clinics, and self-referrals from within the state of Connecticut. Clinic sessions are the first and third Wednesday afternoon each month. The patient visit, any additional required diagnostic testing and/or medications are provided at no cost as any balance remaining after insurance is covered by the Winchester Fund of Yale New Haven Health. The majority of patients are referred by pediatric offices and are international families whose children have travelled to or emigrated from TB-endemic countries. As an example of the diverse patient population, interpreter services were required for Spanish, Arabic, Korean, Farsi, Swahili, Pashto, Haitian Creole, French, Dari, Mandarin and Laotian.

### 2.2. Study Design

An initial retrospective chart review of pediatric patients aged 0–18 years referred to the Yale Pediatric Winchester Chest Tuberculosis Clinic from 2016 to 2019 was performed. Inclusion criteria were children ages 0–18 years and evaluated in the clinic for TBI. Exclusion criteria were any patient seen >18 years or that was evaluated and treated for active TB disease. Demographic and clinical data were compiled, including age, gender, ethnicity, race, language, referring provider (self-referral, physician referral, immigration office referral, or school office referral), distance to clinic, and treatment type (isoniazid, rifampin, or both/multi-drug). The number of missed appointments was recorded for each patient, as was the number of patients who completed therapy.

The clinic saw very few patients early in the COVID-19 pandemic as most pediatricians were not seeing patients in-person for routine well child visits and screening for TBI. Additionally, travel was reduced and this likely reduced referrals. Pediatric clinics at Yale New Haven Children’s Hospital introduced telehealth widely into different locations to care for well and sick children for both new referrals and follow-up visits. As the pediatric TB clinic volume slowly increased, we adapted telehealth to best meet the needs of our patient population. We developed a strategy where all new patient visits were in-person so a thorough history and physical exam could be performed and treatment reviewed in detail. In an effort to reduce the risk of SARS-CoV-2 exposure, we introduced the option of telehealth visits for follow-up appointments to monitor medication adherence, new symptoms, and adverse events. A follow-up visit was scheduled for every patient one month after starting TBI therapy and monthly visits were continued for most patients on the 9H regimen. For patients on the 4R regimen subsequent follow-up ranged from one–three months depending on whether the child was having any adverse effects of the medication or difficulty with adherence to the medication regimen. Families were given a phone number to call at any time with concerns about adverse events or new symptoms worrisome for TB disease. The final visit was in-person to provide documentation of therapy completion, perform a final physical exam, and answer any questions about future screening and exposure. In-person visits are seen by a pediatric infectious disease fellow and attending and generally lasted 30–45 min for new patients and 15 min for follow-ups. Telehealth visits were performed via Zoom (San Jose, CA, USA) integrated into the electronic health record (EHR) and generally lasted 10–15 min. An interpreter was accessible within the telehealth visit. Patients were required to sign up for a patient portal to access the visit. Instructions for patient portal enrollment were provided at the initial visit and available in English and Spanish or instructions were available with the assistance of an interpreter. Nursing staff in the clinic also called patients the day of the appointment to make sure they were set up appropriately for the telehealth visit. After 1 year of implementation from January 2021 to December 2021, we re-evaluated the no show rate and examined patient factors associated with telehealth visits. Because of the small sample size, in the second cohort we included all patients evaluated for TBI including two who were over 18 years. This involved 29 new patients and 52 total visits.

Missed appointments were defined as a pediatric patient not showing up to an appointment during their treatment window. Cancelled appointments were not included as missed appointments. Patients who did not have a final visit to document taking the appropriate medication in the required time frame were categorized as failure to complete therapy and we did not distinguish between primary and secondary non-adherence [2]. Descriptive statistics included frequencies and percentages. Two types of statistical analyses were performed: (1) generalized estimating equations and (2) bivariate chi-square tests. Generalized estimating equations was used to model repeated visits from the same patients to analyze the associations between demographic/clinical variables and missed appointments. Bivariate chi-square tests were performed to analyze the associations between demographic/clinical variables and missed appointments as well as failure to complete treatment. The study was approved by the Institutional Review Board of Yale University School of Medicine (Protocol # 2000026306 approved 23 October 2019).

## 3. Results

One hundred twenty-nine patients, ages 0–18 years, were evaluated in the Yale Pediatric Winchester Chest Tuberculosis clinic from 2016–2019 prior to the introduction of telehealth. Twenty-nine patients were evaluated after telehealth implementation in 2021 (Table 1). There were no significant differences comparing patient characteristics in each group (Table 1). Most patients identified as Hispanic, male, English-speaking, were >12 years, and were referred from a private office (Table 1). As daily rifampin has been endorsed as a preferred therapy to daily isoniazid for eligible patients, more patients were prescribed 4R therapy in 2021 compared with those treated in 2016–2019 (*p* < 0.001) who predominately received 9H (Table 1).

### 3.1. Missed Appointments before Telehealth Implementation

Of 129 patients seen from 2016–2019, 91 (70.5%) patients were initially diagnosed with TBI or exposed to a case of active TB disease and prescribed medication for treatment. More than half of the patients 47/91 (51.6%) missed at least one appointment and 21/91 (23.1%) missed two or more in-person appointments. Four hundred seventy-four appointments were scheduled and not cancelled with 80/474 (16.9%) missed appointments (Table 2). Primary language was significantly associated with missed appointments (*p* = 0.003), with the highest proportion of missed appointments among patients speaking a primary language other than English or Spanish (32.8%) (Table 2). These languages included Arabic, Korean, Farsi, Swahili, Pashto, Haitian Creole, French, Dari, Mandarin and Laotian. Race/ethnicity was significantly associated with missed appointments, as patients identifying as Non-Hispanic Black (27.4%) or Non-Hispanic Other (27.3%) missed the most appointments (Table 2). Surprisingly, patients living within 0–5 miles of the clinic accounted for the greatest proportion of missed appointments (23.2%) (*p* = 0.002) (Table 2). Patients referred by a primary care provider were less likely to miss appointments (*p* = 0.033).

### 3.2. Incomplete Therapy before Telehealth Implementation

Among 91 patients with TBI or exposed to a case of active TB disease and prescribed medication for treatment, five did not develop TBI on further evaluation and treatment was discontinued. Of the 86 patients (66.7%) diagnosed with TBI and treated, 62 (72.1%) completed therapy. A higher proportion (47.6%) of Spanish-speaking patients were more likely to not complete therapy (Table 3) (*p* = 0.031). There was an association between the number of missed appointments and failure to complete treatment (*p* = 0.050) (Table 3). Of the 44 patients who missed zero appointments, only 7 (15.9%) failed to complete therapy (Table 3). Of the 47 patients who missed one or more appointments, 17 (36.2%) failed to complete therapy, and of the 21 patients who missed two or more appointments, 9 (42.9%) failed to complete therapy (Table 3). Distance to clinic (*p* = 0.029) and medication prescribed (*p* = 0.027) were also significantly associated with not completing TBI therapy, such that patients who lived closest or farthest away and those prescribed rifampin were less likely to complete therapy (Table 3).

### 3.3. Missed Appointments after Telehealth Implementation

As the COVID-19 pandemic continued, telehealth was introduced as an option for follow-up visits. From January–December 2021, there were a total of 52 visits from 29 unique patients. There were 5.8% (3/52) missed appointments during 2021 regardless of visit type, which was significantly lower compared with the earlier cohort, that had 16.9% (80/474) missed appointments (*p* = 0.037). Of all follow-up visits, 54.2% (13/24) were telehealth. The proportion of missed in-person appointments (5.3%, 2/38) did not differ significantly from the proportion of missed telehealth visits (7.1%, 1/14) (*p* = 0.818). One new patient visit was performed via telehealth due to transportation barriers preventing an in-person intake appointment. We are not aware of any adverse events related to patients seen via telehealth. There were no differences in the characteristics of patients who chose telehealth for follow-up instead of in-person except that being further from the clinic increased the probability of choosing a telehealth follow-up (2/19 = 11.0% < 10 miles vs. 12/33 = 36.4% > 10 miles).

### 3.4. Incomplete Therapy after Telehealth Implementation

Among 29 patients with TBI after telehealth implementation, there were 19 patients (65.5%) with TBI or exposed to a case of active TB disease and prescribed medication for treatment. Of these 19 patients, 7 patients (36.8%) completed therapy, 1 patient (5.3%) failed to complete therapy, and 11 patients (57.9%) are still undergoing therapy. Due to the high proportion of subjects still undergoing therapy and small sample size, the effect of telehealth on TBI treatment completion remains to be determined.

## 4. Discussion

This retrospective chart review study evaluated factors associated with missed appointments and TBI treatment completion in a free pediatric TB clinic before and during the COVID-19 pandemic when telehealth was introduced as an option for follow-up visits. VDOT has long been studied for adherence to active TB therapy and studies find that it is feasible, acceptable, and cost-effective [18,19,20,21]. For TBI treatment, telehealth has been used to connect TB experts to local pediatricians for patient discussions to reduce the need for patient travel [22]. This resulted in decreased time to treatment completion and an increased percentage of children diagnosed prior to the onset of TB symptoms [22].

Our current study demonstrated that language and lack of a referring primary care provider were significantly associated with missed appointments in pediatric TBI patients prior to the COVID-19 pandemic. These findings highlight the importance of reliable language interpreter services and interventions to improve communication and follow-up, especially with non-English speaking patients. Previous studies identified that lack of knowledge and risk perception, as well as inadequate programmatic interventions to optimize follow-up, were key factors in non-adherence and these were not evaluated in this study [23,24,25]. Acknowledging the need for more effective communication and implementing interventions, such as providing in-person or virtual interpreters and printing out information in the patient’s native language, may reduce missed appointments. Similarly, identifying children not referred by a primary care provider as high risk for missed appointments and targeting this group for additional communication to ensure follow-up may help increase adherence to TBI therapy. TB clinics often serve as an entrance into the US healthcare system for immigrants and refugees, and thus they should have resources available to link families with pediatric clinics to optimize care and perform a thorough health assessment.

It is unclear why from 2016–2019 the majority of children who missed appointments were within five miles of the clinic. Possible reasons for this observation include trouble accessing transportation within the immediate towns surrounding the clinic or difficulty attending a clinic that meets only on Wednesday afternoons twice monthly. During the COVID-19 pandemic, telehealth was utilized most by patients who had to travel further for the visit and those closer to the clinic attended in-person. While it is not surprising that patients who travel further to reach the clinic prefer telehealth, it highlights the value of video visits for referral specialty clinics that meet infrequently and in a single location. The lack of travel associated with telehealth also allows parents/caregivers working from home to continue working until the provider is ready to see the patient, essentially eliminating wait times. It is important to note that the number of missed appointments for in-person visits also declined in 2021 compared with 2016–2019. We hypothesize that patients with difficulty attending in-person follow-up self-selected the telehealth option and only patients who knew they could attend in-person chose this option. This would explain why in-person no show rates declined when telehealth was introduced. Alternatively, this cohort may have been more interested in being seen in-person despite the risks of SARS-CoV-2 exposure or have a more flexible work schedule due to changing attitudes towards working from home during the pandemic.

One concern regarding telehealth is the availability of the required devices to access the EHR remotely. We found that families had access to a cell phone and therefore were able to participate in telehealth. The additional communication provided by nursing to ensure enrollment in the EHR patient portal also served as crucial to delivering appointment reminders and likely contributed to improved patient compliance with these visits. Our initial goal was that all new visits be in-person. However, as comfort with telehealth increased, we now consider telehealth visits for families that have great difficulty traveling to the clinic when there is documentation of the positive TB screening diagnostic test and a negative chest X-ray. This flexibility allows patients to be evaluated and treated that likely would not have shown up for their clinic appointment or perhaps not even scheduled an initial visit.

Limitations include that this is a single center study with a relatively small sample size, especially in the second cohort where many patients are still receiving therapy such that we could not discern whether telehealth facilitated treatment completion. The reasons for missed appointments were not identified. Patients seeking care during the pandemic may be different in that they were more interested in being treated for TBI.

The successful management and treatment of pediatric patients with TBI is of paramount importance in ending the TB epidemic in the United States and globally, especially during the COVID-19 pandemic [11]. The pandemic has exacerbated the TB epidemic, as for the first time since 2005, the World Health Organization (WHO) has seen a year-over-year increase in global TB mortality caused primarily by decreased access to care [11]. Other pandemic-related setbacks include a 15% reduction in the number of patients treated for TB and a 21% decrease in individuals receiving preventative treatment for TBI [11]. Therefore, it is more urgent than ever to address the social determinants of TBI treatment incompletion and consider options such as increased telehealth services to reduce barriers to TBI care.

## 5. Conclusions

This study demonstrates that missed appointments prior to the COVID-19 pandemic were higher in patients who spoke languages other than Spanish and English, were not referred by a primary care provider and who lived within five miles of the clinic. These children may benefit from targeted communication to facilitate adherence to TBI care. The introduction of telehealth reduced no-show rates for both video and in-person visits and, in this small sample, was a valuable tool for TBI follow-up and was favored by patients who lived further from the clinic.

## Figures and Tables

**Table 1 tropicalmed-07-00026-t001:** Study characteristics of pediatric patients prior to and after telehealth (2016–2019, 2021).

Characteristic	Prior to Telehealth (2016–2019) (N = 129), No. (%)	After Telehealth (2021) (N = 29), No. (%)	*p*-Value *
Race/Ethnicity			0.207
NH ^#^ White	14 (10.9)	0 (0.0)	
NH Black	17 (13.2)	3 (10.3)	
NH Asian	25 (19.4)	10 (34.5)	
Hispanic	56 (43.4)	13 (44.8)	
NH Other	17 (13.2)	3 (10.3)	
Gender, Female	58 (45.0)	13 (44.8)	0.990
Language			0.179
English	81 (62.8)	13 (44.8)	
Spanish	33 (25.6)	10 (34.5)	
Other	15 (11.6)	6 (20.7)	
Distance to clinic, miles			0.154
0–5	69 (53.5)	9 (31.0)	
5–10	19 (14.7)	5 (17.2)	
10–20	18 (14.0)	6 (20.7)	
>20	23 (17.8)	9 (31.0)	
Medication			<0.001
Isoniazid (9H)	67 (51.9)	0 (0.0)	
Rifampin (4R)	15 (11.6)	18 (62.1)	
Both/Multi-drug ^	9 (7.0)	1 (3.5)	
No Medication	38 (29.5)	10 (34.5)	
Referring provider			0.766
Private Office	99 (76.7)	23 (79.3)	
Self-referral/Other	30 (23.3)	6 (20.7)	
Age Group (3 Categories)			0.183
<6 years	28 (21.7)	2 (6.9)	
6–12 years	47 (36.4)	13 (44.8)	
>12 years	54 (41.9)	14 (48.3)	

* Denotes χ2 test. ^#^ Non-Hispanic. ^ H or R, with medications switched due to adverse drug reactions.

**Table 2 tropicalmed-07-00026-t002:** Study characteristics and missed appointments among total appointments for patients initially diagnosed with TBI or exposed to a case of active TB disease prior to telehealth (2016–2019). N = 474.

Characteristic	No. of Appointments	No. (%) of Missed Appointments, (N = 80)	*p*-Value
Race/Ethnicity			0.003 *
NH ^#^ White	45	8 (17.8)	
NH Black	62	17 (27.4)	
NH Asian	108	9 (8.3)	
Hispanic	193	28 (14.5)	
NH Other	66	18 (27.3)	
Gender			0.347
Male	238	44 (18.5)	
Female	236	36 (15.3)	
Language			0.003 *
English	320	46 (14.4)	
Spanish	96	15 (15.6)	
Other	58	19 (32.8)	
Distance to clinic, miles			0.002 *
0–5	211	49 (23.2)	
5–10	101	17 (16.8)	
10–20	69	4 (5.8)	
>20	93	10 (10.8)	
Medication			0.032 *
Isoniazid (9H)	316	55 (17.4)	
Rifampin (4R)	36	11 (30.6)	
Both/Multi-drug ^	66	10 (15.2)	
No Medication	56	4 (7.1)	
Referring provider			0.033 *
Private Office	353	52 (14.7)	
Self-referral/Other	121	28 (23.1)	
Age Group (3 Categories)			0.826
<6 years	96	18 (18.8)	
6–12 years	177	28 (15.8)	
>12 years	201	34 (16.9)	

* Denotes significance < 0.05 for χ2 test. ^#^ Non-Hispanic. ^ H or R, with medications switched due to adverse drug reactions.

**Table 3 tropicalmed-07-00026-t003:** Study characteristics and failure to complete therapy of TBI among patients initially diagnosed with TBI or exposed to a case of active TB disease and prescribed medication for treatment prior to telehealth (2016–2019). N = 91.

Characteristic	No.	No. (%) Failure to Complete Therapy, (N = 24)	*p*-Value
Missed Appointments			0.050
0	44	7 (15.9)	
1	26	8 (30.8)	
2	13	7 (53.8)	
3+	8	2 (25.0)	
Race/Ethnicity			0.147
NH ^#^ White	7	1 (14.3)	
NH Black	13	4 (30.8)	
NH Asian	18	2 (11.1)	
Hispanic	39	15 (38.5)	
NH Other	14	2 (14.3)	
Gender, Female	42	9 (21.4)	0.322
Language			0.031 *
English	60	13 (21.7)	
Spanish	21	10 (47.6)	
Other	10	1 (10.0)	
Distance to clinic, miles			0.029 *
0–5	46	16 (34.8)	
5–10	17	0 (0.0)	
10–20	15	3 (20.0)	
>20	13	5 (38.5)	
Medication			0.027 *
Isoniazid (9H)	67	15 (22.4)	
Rifampin (4R)	15	8 (53.3)	
Both/Multi-drug ^	9	1 (11.1)	
Referring provider			0.409
Private Office	70	17 (24.3)	
Self-referral/Other	21	7 (33.3)	
Age Group (3 Categories)			0.104
<6 years	17	1 (5.9)	
6–12 years	36	11 (30.6)	
>12 years	38	12 (31.6)	

* Denotes significance < 0.05 for χ2 test. ^#^ Non-Hispanic. ^ H or R, with medications switched due to adverse drug reactions.

## Data Availability

Data available on request due to privacy restrictions. De-identified data presented in this study are available on request from the corresponding author. The data are not publicly available due to privacy concerns regarding patient data.

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
