# Peer review of "Telehealth Reduces Missed Appointments in Pediatric Patients with Tuberculosis Infection"

_tropicalmed, 2022, doi:10.3390/tropicalmed7020026_

Round 1

Reviewer 1 Report

A well-written manuscript that is timely given the ongoing pandemic.

Only a few minor comments for clarification

1) Methods: Line 104-105 is this a one-year period or two  as text and dates do not match-  Please check January 2020 to December 2021 

2) Please clearly identify the TB regime used in the evaluated cohort and the previous cohort - was it 3HP and 6H -- The regimens should be clearly identified using recognized acronyms.  

3) what was the frequency of follow up visits for both cohort 

4) on average how long does a clinic visit last -  would parents need a day off from work? I am wondering if that played a role in parents not being able to attend the appointment in the previous cohort as compared to during the pandemic given work from home. 

Reviewer 2 Report

The title of the article looks like a sentence and OR maybe a conclusion; some rephrasing is suggested. While feel free either accept or reject the reviewer proposed title like;

Missed Appointments in Pediatric Patients with Tuberculosis Infection and Impact of Telehealth

Abstract:

  1. The first sentence; “Pediatric patients with untreated tuberculosis infection (TBI) are at risk of progression to active TB disease” sound is not clear. Does the author mean latent TB or active TB but untreated? It is suggested to rephrase.

Introduction:

  1. I will suggest adding telehealth impact on adherence level and or on outcomes of TB patients in the introduction section, as the author mentioned and attempt to link at page 3, 2nd line in methodology.
  2. Study Setting detail information is suggested because provided detail might be enough for the local readers but not for global perspective (location and or population covered by the cited hospital?).
  3. Study Design; Author mentioned that due to the pandemic pediatricians were not seeing patients in person & screening for TBI. Researchers developed a new strategy to overcome this challenge. It will appreciate it if the author(s) provide some detail. Further, these developed strategies were adopted/practiced in cited clinic/center OR generalised in other health clinics as well. It is interesting.
  4. Methodology section; Last paragraph (page 3 of PDF file); Some definitions were observed which are standard, but it is appreciated if the author provides a reference.
  5. Ethical approval should mention in the methodology section of the article.
  6. Abbreviations either in the tables or in the text shouldn’t be placed earlier and proper description is advised. In tables, the footnote is suggested.
  7. Inclusion & Exclusion criteria need to describe enough in detail.
  8. Treatment outcomes were not mentioned among the telehealth group, so how should we assume how much it is effective. That’s why criteria, sample size and limitations need to rephrase.
  9.  
